# Bacterial Competition in the Presence of a Virus in a Chemostat

**Amer Hassan Albargi** [1] and **Miled El Hajji** [2,3,*]

1   Department of Mathematics, Faculty of Science, King Abdulaziz University, P.O. Box 80203,
    Jeddah 21589, Saudi Arabia; aalbarqi@kau.edu.sa
2   ENIT-LAMSIN, Tunis El Manar University, BP. 37, Tunis-Belvédère, Tunis 1002, Tunisia
3   Department of Mathematics, Faculty of Science, University of Jeddah, P.O. Box 80327,
    Jeddah 21589, Saudi Arabia
*   Correspondence: miled.elhajji@enit.rnu.tn

**Abstract:** We derive a mathematical model that describes the competition of two populations in a chemostat in the presence of a virus. We suppose that only one population is affected by the virus. We also suppose that the substrate is continuously added to the bioreactor. We obtain a model taking the form of an "SI" epidemic model using general increasing growth rates of bacteria on the substrate and a general increasing incidence rate for the viral infection. The stability of the steady states was carried out. The system can have multiple steady states with which we can determine the necessary and sufficient conditions for both existence and local stability. We exclude the possibility of periodic orbits and we prove the uniform persistence of both species. Finally, we give some numerical simulations that validate the obtained results.

**Keywords:** chemostat; competition; virus; coexistence; local stability; uniform persistence

**MSC:** 34C60; 34C23; 92D25; 93D30; 93D20; 34C37; 34C55; 34C15

## 1. Introduction

A bioreactor (Figure 1) is a tank (a lake) in which microorganisms multiply (yeasts, bacteria, fungi, algae, etc.) which consume substrates or feed on other organisms to develop, and which use precursors and activators to produce biomass, synthesize metabolites, or even bioconvert molecules of interest (e.g., depollution). Thanks to the bioreactor, it is possible to control the culture conditions (temperature, pH, aeration, etc.) and, therefore, to collect relatively reliable experimental data for monitoring bacterial growth and/or the chemical reaction of interest. If we consider a competition between two species for an essential substrate, a classical postulate, known as the competitive exclusion principle, suggests that, at most, one species can survive and the other species disappear. This principle has been frequently demonstrated mathematically and validated experimentally (see, for example, [1–5]). Several works [6–14] have tried to explain coexistence of bacterial competitors using several approaches. Increasingly interested in aquatic environments, researchers are discovering that the organisms colonized by viruses are much more varied than the bacterial species anticipated [15]. The shape and size of some viruses are also surprising. Finally, we are beginning to measure the impact of viral diversity on the living world. Through various mechanisms, such as the destruction of a dominant species to the benefit of rarer species [16] or the transfer of viral genes to the host, viruses (bacteriophage) maintain the biodiversity of aquatic ecosystems and facilitate genetic mixing [17]. Several works [17,18] confirm that viruses have a significant role in aquatic bacterial diversity. Therefore, the role of viruses in aquatic ecosystems cannot be neglected and should be taken into account when modeling bacterial competition in an aquatic ecosystem.

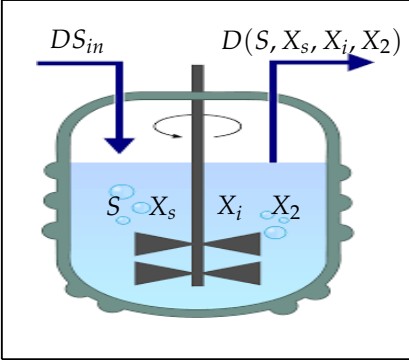

**Figure 1.** A chemostat is a well-stirred bioreactor [19] where a limiting substrate ($S_{in}$) is continuously added to a liquid culture containing two competitors ($X_s$, $X_i$, $X_2$) in the presence of a virus affecting only the first competitor.

Note that a viral infection can be modeled with an epidemiological model, using either a deterministic, delayed, or stochastic approach [20–24]. One of the basic models for the spread of a disease was proposed in [25], dividing the population into three compartments, namely the infected population compartment ($I$), the susceptible population compartment ($S$), and the recovered population compartment ($R$), known as the "SIR" model. An extension of the "SIR" models is given by the "SEIR" (Susceptible, Exposed, Infected, Recovered) ones [26–29]. The "SEIR" epidemic models were extended to "SVEIR" models (Susceptible, Vaccinated, Exposed, Infected, Recovered), taking into account the proportion of immigrants who have been vaccinated [30–33]. Most of these works investigated the proposed models by giving the basic reproduction number and the local and global stability of the steady states using local linearisation and Lyapunov theory.

An important question has been asked in [34]: does the presence of a virus induce the stable coexistence of bacterial competitors in an aquatic-like system?

The response was given by proposing and analyzing a mathematical model of exploitative competition in a continuous reactor containing a virus [34]. The authors assume that only the species which have the best affinity with the substrate are affected by the virus. They proved under certain conditions that the coexistence of competitor bacteria is possible. Mestivier et al. [35] and Weitz et al. [36] proposed some mathematical models where the virus dynamics are given explicitly. It is shown that the coexistence between two competitor bacteria is possible in the presence of a virulent virus. Similarly, in [37], the authors considered a mathematical model where the virus behaviour is given explicitly and they give some conditions satisfying the coexistence of all competitors.

In this paper, we propose a generalized model of the one given in [34] by considering general increasing growth rates of bacteria on the substrate and a general increasing incidence rate for the viral infection. We introduce the model in Section 2 and we give some general results. In Section 3, we discuss the case where there is no viral infection where the competitive exclusion principle is valid. In Section 4, we reduce the system to a three-dimensional one which facilitates the mathematical analysis. We discuss the local analysis in Section 4.1, we prove that there is no periodic orbits on the faces in Section 4.2, and then we conclude on the persistence in Section 4.3 and the uniform persistence in Section 4.4. Then, we return in Section 5 to the main model where we discuss the uniform persistence. Then, we give some numerical simulations in Section 6. Finally, we summarize the main results and discuss certain implications in Section 7.

## 2. Modeling Bacterial Competition in the Presence of a Virus

Consider a bio-reactor in which the bacterial competition of two species in the presence of a virus that affects only species 1 was studied (see Figures 1 and 2). Therefore, species 1 is present in two compartments, susceptibles ($X_s$) and infectives or bacteriophage ($X_i$); however, species 2 is present in a single form ($X_2$). We know that a virus requires a host

to replicate; that is why we do not explicitly model the virus dynamics and we assume that the virus spreads when an infected species takes contact with a susceptible one, which is the case of the classical epidemic models ("SI", "SIS", "SIR", "SIRS", "SEIR", "SEIRS", "SVEIR"). The limiting substrate ($S$) was added instantaneously to the reactor with a flow rate $D$ and a concentration $S_{in}$. The culture liquid, containing the substrate, species 1(either infected or not), and species 2, is continuously mixed and removed at the same flow rate, $D$. Note that the viral infection concerns only species 1. We neglected all natural mortality rates compared to the dilution rate.

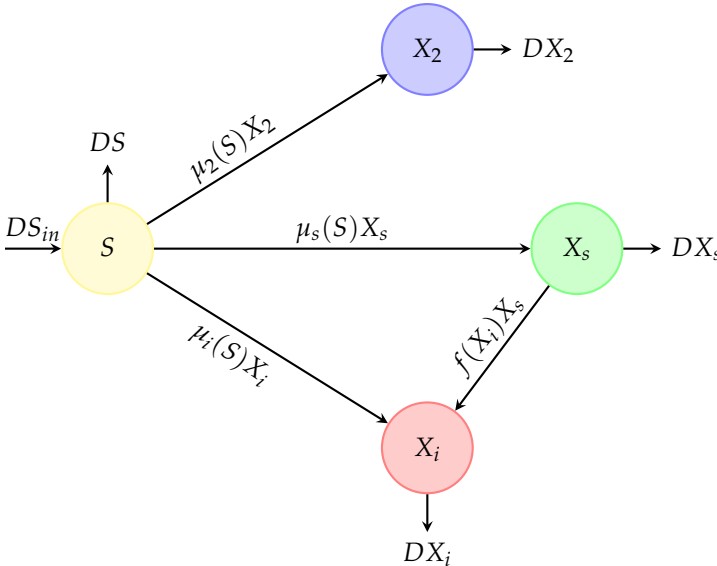

**Figure 2.** Competition diagram of the competition of the two species in the presence of a virus inside a bioreactor. Compartments $S$, $X_s$, $X_i$, and $X_2$ are described by circles and transition rates between compartments are described by arrows and labels.

We proposed a mathematical model describing the competition of two species for a single non-reproducing growth-limiting substrate in a continuous reactor that it is well-stirred in the presence of a virus that affects only species 1 (species 2 is not susceptible to the virus attack, Figure 2). The mathematical model takes the form of an "SI" epidemic model where the main goal is to find under what conditions the coexistence of all species is possible. This model is a generalization of the model proposed in [34] by considering generalized growth rates for all species and also a generalized incidence rate for the viral infection. The model is given by the following fourth-dimensional system of ordinary differential equations:

$$
\begin{cases}
\dot{S} &= D(S_{in} - S) - \dfrac{\mu_s(S)}{Y_1}X_s - \dfrac{\mu_i(S)}{Y_1}X_i - \dfrac{\mu_2(S)}{Y_2}X_2, \\
\dot{X}_s &= \mu_s(S)X_s - DX_s - f(X_i)X_s, \\
\dot{X}_i &= \mu_i(S)X_i - DX_i + f(X_i)X_s, \\
\dot{X}_2 &= \mu_2(S)X_2 - DX_2.
\end{cases}
\tag{1}
$$

Here, $S$ denotes the concentration of the resource with $S(0) \geq 0$, whereas $X_s$, $X_i$, and $X_2$ stand for the concentrations of susceptible species 1, infected species 1, and species 2, respectively, with initial conditions satisfying $X_s(0) > 0$, $X_i(0) > 0$ and $X_2(0) > 0$. Note that $D$ and $S_{in}$ describe the dilution rate and the substrate input concentration, respectively, and are assumed to be constant and positive. $Y_1$ and $Y_2$ denote the yield coefficients, commonly referred to as the substrate-to-species-1 (either infected or not) and substrate-to-species-2 yields, respectively. The significance of the variables and parameters is shown in Table 1.

**Table 1.** Variables and parameters meaning of system (1).

| Notation | Description |
|---|---|
| $\mu_s(\cdot)$ | Specific growth rate of susceptible species 1 |
| $\mu_i(\cdot)$ | Specific growth rate of infected species 1 |
| $\mu_2(\cdot)$ | Specific growth rate of species 2 |
| $f(\cdot)$ | Saturated incidence rate |
| $S_{in}$ | Substrate input concentration |
| $D$ | Flow rate |
| $Y_1$ | Yield coefficient expressing the substrate-to-species-1 yield |
| $Y_2$ | Yield coefficient expressing the substrate-to-species-2 yield |

By making the following change of variable, we obtain a more simplified model. Let $s = S, s_{in} = S_{in}, x_s = \dfrac{X_s}{Y_1}, x_i = \dfrac{X_i}{Y_1}, x_2 = \dfrac{X_2}{Y_2}$, and $\mu(x_i) = f(Y_1 x_i)$. Then, the model takes the form:

$$\begin{cases} \dot{s} &= D(s_{in} - s) - \mu_s(s)x_s - \mu_i(s)x_i - \mu_2(s)x_2, \\ \dot{x}_s &= \mu_s(s)x_s - Dx_s - \mu(x_i)x_s, \\ \dot{x}_i &= \mu_i(s)x_i - Dx_i + \mu(x_i)x_s, \\ \dot{x}_2 &= \mu_2(s)x_2 - Dx_2. \end{cases} \tag{2}$$

Let us define some operating parameters as follows: $D_s = \mu_s(s_{in})$, $D_2 = \mu_2(s_{in})$, and $D_i = \mu_i(s_{in})$. All the mentioned parameters are positive. Through the paper, we will consider the most important case by using the following assumption:

**Assumption 1.** *The growth rates $\mu_s$, $\mu_i$, and $\mu_2$ are increasing, non-negative, $C^1(\mathbb{R}_+)$ functions, such that $\mu_s(0) = \mu_i(0) = \mu_2(0) = 0$. Furthermore, $\mu_i(s) < \mu_2(s) < \mu_s(s)$ for all $s \in (0, s_{in})$ and $D_i < D < D_2$.*

Let us define the values $\bar{s}_1$ and $\bar{s}_2$ as the solutions of $\mu_s(s) = D$ and $\mu_2(s) = D$, respectively.

**Remark 1.**

1. *Assumption 1 expresses that species 1 has the best affinity with the substrate and then it wins the competition in the absence of the infection. Once the infection is present, Assumption 1 expresses that the non-infected species 1 ($x_s$) still has the best affinity with the substrate; however, infected species 1 ($x_i$) has a growth rate ($\mu_i$) smaller than both growth rates ($\mu_s$ and $\mu_2$) of the non-infected species 1 and species 2.*
2. *Monod functions (or Holling's functions type II) are candidate functions that can express growth rates (Figure 3):*

$$\mu_s(s) = \frac{\bar{\mu}_s s}{k_s + s}, \mu_i(s) = \frac{\bar{\mu}_i s}{k_i + s}, \mu_2(s) = \frac{\bar{\mu}_2 s}{k_2 + s} \text{ and } \mu(s) = \frac{\bar{\mu} s}{k + s}$$

*where $k_s, k_i$, and $k_2$ are Monod constants. $\bar{\mu}_s, \bar{\mu}_i$, and $\bar{\mu}_2$ are positive constants. All constants can be chosen such that the functions $\mu_s$, $\mu_i$, and $\mu_2$ satisfy Assumption 1. For example, we can take $\bar{\mu}_s > \bar{\mu}_2 > \bar{\mu}_i$ and $k_s = k_2 = k_i$.*

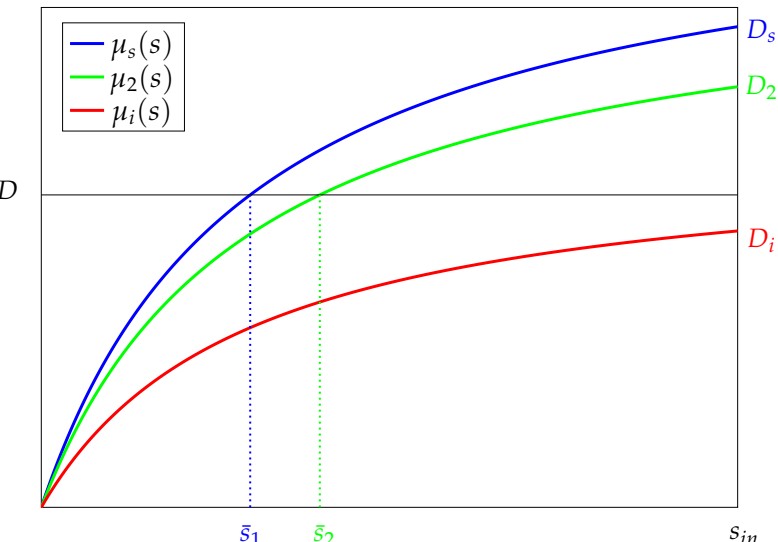

**Figure 3.** Typical growth rates $\mu_s(\cdot)$, $\mu_i(\cdot)$ and $\mu_2(\cdot)$ where $\bar{s}_1 < \bar{s}_2 < s_{in}$ and $D_i < D < D_2 < D_s$.

The model (2) of the chemostat is a dynamical system defined on the non-negative cone, for which we recall some fundamental properties (see, for instance, [38]).

**Proposition 1.** *System (2) satisfies*

1.　*All solutions of system (2) are defined, non-negative, and bounded.*
2.　$\Sigma = \left\{(s, x_s, x_i, x_2) \in \mathbb{R}_+^4 \mid s + x_s + x_i + x_2 = s_{in}\right\}$ *is a positively invariant attractor set of solutions of the system (2).*

**Proof.**

1.　The invariance of $\mathbb{R}_+^4$ is confirmed by the following points: $s(t) = 0 \Rightarrow \dot{s}(t) = Ds_{in} > 0$, $x_i(t) = 0 \Rightarrow \dot{x}_i(t) = 0$, $x_s(t) = 0 \Rightarrow \dot{x}_s(t) = 0$, and $x_2(t) = 0 \Rightarrow \dot{x}_2(t) = 0$.
　　Consider the variable $M(t) = s(t) + x_s(t) + x_i(t) + x_2(t) - s_{in}$. By adding all equations of model (2), we deduce that:

$$\dot{M}(t) = -DM(t), \tag{3}$$

　　and, therefore, we obtain:

$$s(t) + x_s(t) + x_i(t) + x_2(t) = s_{in} + M_0 e^{-Dt}$$

　　with $M_0 = s(0) + x_s(0) + x_i(0) + x_2(0) - s_{in}$. Since all compartments of the sum are non-negative, we can conclude on the boundedness of the solution.
2.　It can be deduced from the relation (3). □

**Lemma 1.** $\bar{s}_1$ *and* $\bar{s}_2$ *exist and are unique and satisfy* $0 < \bar{s}_1 < \bar{s}_2 < s_{in}$.

**Proof.** The function $\mu_s$ is continuous and increasing, such that $\mu_s(0) = 0$ and $D < D_s = \mu_s(s_{in})$; therefore, $\bar{s}_1 \in (0, s_{in})$ exists and is unique. The function $\mu_2$ is continuous and increasing, such that $\mu_2(0) = 0$ and $D < D_2 = \mu_2(s_{in})$; therefore, $\bar{s}_2 \in (0, s_{in})$ exists and is unique. Since $\mu_2(s) < \mu_s(s)$ for all $s \in (0, s_{in})$ then $\bar{s}_1 < \bar{s}_2 < s_{in}$. □

**Assumption 2.** *The incidence rate* $\mu$ *is an increasing, non-negative,* $C^1(\mathbb{R}_+)$ *concave function, such that* $\mu(0) = 0$. *Furthermore,* $\mu$ *satisfies:*

$$\mu_s(\bar{s}_2) < D + \mu(s_{in} - \bar{s}_2) \tag{4}$$

*and*

$$\mu_i'(0) < \mu'(x_i) \quad \forall \ x_i \ \in \mathbb{R}_+. \tag{5}$$

**Remark 2.** *Monod (or Holding type II) function is a candidate function that can express the incidence rate* $\mu(x) = \dfrac{\bar{\mu}x}{k+x}$, *where k is the Monod constant.* $\bar{\mu}$ *is the maximum incidence rate. Note that* $\mu$ *satisfies Assumption* 2.

The incidence rate $\mu$ satisfies the following lemma.

**Lemma 2.** *The incidence rate* $\mu$ *satisfies* $\mu'(x) \leq \dfrac{\mu(x)}{x} \leq \mu'(0), \ \forall x > 0.$

**Proof.** Let $x, x_1 \in \mathbb{R}_+$, and the function $\varphi_1(x) = \mu(x) - x\mu'(x)$. Since $\mu'(x) \geq 0$ ($\mu$ is an increasing function) and $\mu''(x) \leq 0$ ($\mu$ is a concave function), then $\varphi_1'(x) = -x\mu''(x) \geq 0$ and $\varphi_1(x) \geq \varphi_1(0) = 0$. Therefore, $\mu(x) \geq x\mu'(x)$. Similarly, let $\varphi_2(x) = \mu(x) - x\mu'(0)$; then, $\varphi_2'(x) = \mu'(x) - \mu'(0) \leq 0$ once $\mu$ is a concave function. Thus, $\varphi_2(x) \leq \varphi_2(0) = 0$ and $\mu(x) \leq x\mu'(0)$. $\square$

Let us define the basic reproduction number $\mathcal{R}_0$ for system (2) using the next-generation operator approach proposed in [39] and deduced from the third equation (infected compartment) of system (2) and, therefore, given by:

$$\mathcal{R}_0 \quad = \quad \frac{\mu_i(\bar{s}_1) + \mu'(0)(s_{in} - \bar{s}_1)}{D}.$$

Here, $\mu_i(\bar{s}_1) + \mu'(0)(s_{in} - \bar{s}_1)$ describes the mean number of infective produced in a chemostat by introducing a single infective into a totally susceptible population inside the reactor. $\dfrac{1}{D}$ describes the average time that an infective individual passes inside the chemostat as an infective.

For the rest of the paper, we consider the most important case where $\mathcal{R}_0 > 1$.

**Assumption 3.** $\mathcal{R}_0 > 1$ *or, equivalently,* $s_{in} > \dfrac{D + \mu'(0)\bar{s}_1 - \mu_i(\bar{s}_1)}{\mu'(0)}.$

Let us recall the classical 'chemostat' model in the absence of the virus.

## 3. Virus-Free Subsystem

Consider the following three-dimensional system which is the virus-free subsystem:

$$\begin{cases} \dot{s} &= D(s_{in} - s) - \mu_s(s)x_s - \mu_2(s)x_2, \\ \dot{x}_s &= \mu_s(s)x_s - Dx_s, \\ \dot{x}_2 &= \mu_2(s)x_2 - Dx_2. \end{cases} \tag{6}$$

This model is the same as (2) in the absence of the viral infection ($x_i = 0$). This model predicts the competitive exclusion; that is, under Assumption 1, at most, the first species (which has the best affinity with the substrate) avoids extinction; however, the second species goes to extinction (see, for example, [19,38,40,41]). Let us define the steady-states of system (6) on the non-negative quadrant by $SS_0$, $SS_1$, and $SS_2$ with :

$$SS_0 = (s_{in}, 0, 0), \ SS_1 = (\bar{s}_1, s_{in} - \bar{s}_1, 0), \ SS_2 = (\bar{s}_2, 0, s_{in} - \bar{s}_2)$$

where $\bar{s}_1 < \bar{s}_2$ (according to Lemma 1). Therefore, we have:

**Proposition 2.** *The equilibrium point* $SS_1$ *is globally asymptotically stable* [38].

Note that by introducing a virus that affects only species 1, which has the best affinity with the nutriment, we aim to give a possibility of the coexistence of both competing species.

## 4. Reduction to Three-Dimensional System

Note that all solutions of the 4D-dynamics (2) converge toward $\Sigma$. Now, because we are interested by the asymptotic behavior of the dynamics (2), we will restrict the study to $\Sigma$. Thanks to Thieme's results [42], the asymptotic behavior of the reduced dynamics will be informative for the dynamics (2); see [19,43] for other applications. The reduced dynamics of (2) on $\Sigma$ is given by:

$$\begin{cases} \dot{x}_s &= \mu_s(s_{in} - x_s - x_i - x_2)x_s - Dx_s - \mu(x_i)x_s &= x_s\,h_1(x_s, x_i, x_2), \\ \dot{x}_i &= \mu_i(s_{in} - x_s - x_i - x_2)x_i - Dx_i + \mu(x_i)x_s &= x_i\,h_2(x_s, x_i, x_2), \\ \dot{x}_2 &= \mu_2(s_{in} - x_s - x_i - x_2)x_2 - Dx_2 &= x_2\,h_3(x_s, x_i, x_2), \end{cases} \quad (7)$$

where the functions $h_1, h_2$, and $h_3$ are given by:

$$\begin{cases} h_1(x_s, x_i, x_2) &:= \mu_s(s_{in} - x_s - x_i - x_2) - D - \mu(x_i), \\ h_2(x_s, x_i, x_2) &:= \mu_i(s_{in} - x_s - x_i - x_2) - D + \dfrac{\mu(x_i)}{x_i}x_s, \\ h_3(x_s, x_i, x_2) &:= \mu_2(s_{in} - x_s - x_i - x_2) - D. \end{cases} \quad (8)$$

Thus, for (7) the state-vector $(x_s, x_i, x_2)$ belongs to the following subset of $\mathbb{R}^3_+$:

$$\Lambda = \left\{ (x_s, x_i, x_2) \in \mathbb{R}^3_+ : x_s + x_i + x_2 \leq s_{in} \right\}.$$

Formally, let $F_{000}, F_{100}, F_{001}$, and $F_{111}$ be the four equilibrium points of dynamics (7) on $\Lambda$. $F_{000}$ reflects the extinction of all species and predators, and $F_{100}$ reflects the extinction of the infected first species and the second species while the non-infected first species is present. $F_{001}$ reflects the extinction of the first species (either infected or not) while the second species is present. Finally, $F_{111}$ reflects the coexistence of both species including the first species in its two forms, infected or not.
$F_{000}, F_{100}, F_{001}$, and $F_{111}$ are given by:

1. $F_{000} = (0,0,0)$.
2. $F_{100} = (s_{in} - \bar{s}_1, 0, 0)$, where $\bar{s}_1$ is the unique solution of the equation $\mu_s(s) = D$.
3. $F_{001} = (0, 0, s_{in} - \bar{s}_2)$, where $\bar{s}_2$ is the unique solution of the equation $\mu_2(s) = D$.
4. $F_{111} = (\check{x}_s, \check{x}_i, \check{x}_2)$, where $(\check{x}_s, \check{x}_i, \check{x}_2)$ is the solution of the three-dimensional system given by:

$$\begin{cases} h_1(x_s, x_i, x_2) &= \mu_s(s_{in} - x_s - x_i - x_2) - D - \mu(x_i) &= 0, \\ h_2(x_s, x_i, x_2) &= \mu_i(s_{in} - x_s - x_i - x_2) - D + \dfrac{\mu(x_i)}{x_i}x_s &= 0, \\ h_3(x_s, x_i, x_2) &= \mu_2(s_{in} - x_s - x_i - x_2) - D &= 0. \end{cases} \quad (9)$$

From the third equation of system (9), and by Assumption 1, there exists a unique value $\bar{s}_2 \in (0, s_{in})$, such that $\bar{s}_2 = s_{in} - x_s - x_i - x_2$. Thus, $x_2 = s_{in} - x_s - x_i - \bar{s}_2$ and the system (9) is reduced to:

$$\begin{cases} \mu_s(\bar{s}_2) - D - \mu(x_i) &= 0, \\ \mu_i(\bar{s}_2) - D + \dfrac{\mu(x_i)}{x_i}x_s &= 0. \end{cases} \quad (10)$$

From the second equation of (10), we have $x_s = (D - \mu_i(\bar{s}_2))\dfrac{x_i}{\mu(x_i)}$. Let $\varphi_i(x_i) = \mu_s(\bar{s}_2) - D - \mu(x_i)$. Therefore, from the first equation of (10), we have:

$$\varphi_i(x_i) = \mu_s(\bar{s}_2) - D - \mu(x_i) = 0.$$

The derivative of $\varphi_i$ is given by:

$$\varphi_i'(x_i) = -\mu'(x_i) < 0.$$

Furthermore, we have:

$$\begin{array}{rcl} \varphi_i(0) & = & \mu_s(\bar{s}_2) - D > 0, \\ \varphi_i(s_{in} - \bar{s}_2) & = & \mu_s(\bar{s}_2) - D - \mu(s_{in} - \bar{s}_2) \\ & < & 0 \text{ by Assumption 2.} \end{array} \qquad (11)$$

Therefore, the equation $\varphi_i(x_i) = 0$ admits a unique solution $\check{x}_i \in (0, s_{in} - \bar{s}_2)$ and, thus, the existence and uniqueness of the equilibrium point $F_{111}$ corresponding to the coexistence of all species:

$$F_{111} = (\check{x}_s, \check{x}_i, \check{x}_2) = ((D - \mu_i(\bar{s}_2))\frac{\check{x}_i}{\mu(\check{x}_i)}, \check{x}_i, s_{in} - (D - \mu_i(\bar{s}_2))\frac{\check{x}_i}{\mu(\check{x}_i)} - \check{x}_i - \bar{s}_2).$$

The following equilibrium points are either not generic or not possible; that is why they are neglected.

1. $F_{101} = (\tilde{x}_s, 0, \tilde{x}_2)$, where $(\tilde{x}_s, \tilde{x}_2)$ is the solution of the two-dimensional system given by:

$$\begin{cases} h_1(x_s, 0, x_2) & = & \mu_s(s_{in} - x_s - x_2) - D & = & 0, \\ h_3(x_s, 0, x_2) & = & \mu_2(s_{in} - x_s - x_2) - D & = & 0. \end{cases} \qquad (12)$$

This case will be ignored since it is non-generic because we obtain $\bar{s}_1 = \bar{s}_2$ (classical model of bacterial competition in a chemostat).

2. $F_{011} = (0, \hat{x}_i, \hat{x}_2)$, where $(\hat{x}_i, \hat{x}_2)$ is the solution of the two-dimensional system given by:

$$\begin{cases} h_2(0, x_i, x_2) & = & \mu_i(s_{in} - x_i - x_2) - D & = & 0, \\ h_3(0, x_i, x_2) & = & \mu_2(s_{in} - x_i - x_2) - D & = & 0. \end{cases} \qquad (13)$$

This case will be ignored since we obtain $\mu_2(s_{in} - x_i - x_2) = \mu_2(\bar{s}_2) = \mu_i(\bar{s}_2) = D$, which is impossible because $\mu_i(s_{in}) = D_i < D$.

3. $F_{010} = (0, s_{in} - \bar{s}_3, 0)$, where $\bar{s}_3$ is the solution of the equation $\mu_i(s) = D$. Again, this equilibrium is not possible since $\mu_i(s_{in}) = D_i < D$.

4. $F_{110} = (\bar{x}_s, \bar{x}_i, 0)$, where $(\bar{x}_s, \bar{x}_i)$ is the solution of the two-dimensional system given by:

$$\begin{cases} h_1(x_s, x_i, 0) & = & \mu_s(s_{in} - x_s - x_i) - D - \mu(x_i) & = & 0, \\ h_2(x_s, x_i, 0) & = & \mu_i(s_{in} - x_s - x_i) - D + \dfrac{\mu(x_i)}{x_i}x_s & = & 0. \end{cases} \qquad (14)$$

Let:

$$\Gamma_1 = \left\{ (x_s, x_i, 0) \in \Lambda;\ \mu_s(s_{in} - x_s - x_i) - D - \mu(x_i) = 0 \right\}$$

and

$$\Gamma_2 = \left\{ (x_s, x_i, 0) \in \Lambda;\ \mu_i(s_{in} - x_s - x_i) - D + \frac{\mu(x_i)}{x_i}x_s = 0 \right\}.$$

$\Gamma_1$ and $\Gamma_2$ are non-empty and can intersect at a finite number of positive equilibrium points of the form $F_{110} = (\bar{x}_s, \bar{x}_i, 0)$, such that $\bar{x}_s > 0$ and $\bar{x}_i > 0$. Functions $x_s \to \mu_s(s_{in} - x_s - x_i) - D - \mu(x_i)$ and $x_s \to \mu_i(s_{in} - x_s - x_i) - D + \dfrac{\mu(x_i)}{x_i}x_s$ are decreasing. Therefore, the isoclines are the graphs of two functions $x_s = \varphi_1(x_i)$ and $x_s = \varphi_2(x_i)$ and, then, $\varphi_1(0) = s_{in} - \bar{s}_1$ and $\varphi_1(s_{in} - \bar{s}_3) = 0$. $\bar{x}_i$ is solution of $\psi(\bar{x}_i) = 0$, where $\psi(x_i) = \varphi_2(x_i) - \varphi_1(x_i)$. The derivatives of $\varphi_1$ and $\varphi_2$ are given by

$$\varphi_1'(x_i) = -1 - \frac{\mu'(x_i)}{\mu_s'(s_{in} - x_s - x_i)} \text{ and } \varphi_2'(x_i) = -1 + \frac{\left(\mu'(x_i) - \frac{\mu(x_i)}{x_i}\right)\frac{x_s}{x_i} - \frac{\mu(x_i)}{x_i}}{\mu_i'(s_{in} - x_s - x_i) - \frac{\mu(x_i)}{x_i}}.$$

According to Assumption 2, we have $\mu_i'(s_{in} - x_s - x_i) - \frac{\mu(x_i)}{x_i} < 0$, and by Lemma 2,

we have $\mu'(x_i) - \frac{\mu(x_i)}{x_i} \le 0$; therefore, one deduces that:

$$
\begin{aligned}
\psi'(x_i) &= \varphi_2'(x_i) - \varphi_1'(x_i) \\
&= \frac{\mu'(x_i)}{\mu_s'(s_{in} - x_s - x_i)} + \frac{\left(\mu'(x_i) - \frac{\mu(x_i)}{x_i}\right)\frac{x_s}{x_i} - \frac{\mu(x_i)}{x_i}}{\mu_i'(s_{in} - x_s - x_i) - \frac{\mu(x_i)}{x_i}} \\
&> 0.
\end{aligned}
\tag{15}
$$

Note that $\psi(0) = \varphi_2(0) - s_{in} + \bar{s}_1 > 0$, since $\mathcal{R}_0 > 1$ by Assumption 3. Therefore, there is no equilibrium points of the form $F_{110}$ if $D_i < D < D_2$.

Therefore, we will consider only the equilibrium points $F_{000}$, $F_{100}$, $F_{001}$, and $F_{111}$ to be the four equilibrium points of dynamics (7) on $\Lambda$ and we resume them in Proposition 3.

**Proposition 3.** *Under Assumptions 1–3, the dynamics (7) admit four equilibrium points $F_{000}$, $F_{100}$, $F_{001}$, and $F_{111}$.*

### 4.1. Local Stability

The Jacobian matrix at a point $(x_s, x_i, x_2)$ solution of system (7) is given by:

$$J(x_s, x_i, x_2) = \begin{pmatrix} -\mu_s' x_s + \mu_s - D - \mu(x_i) & -\mu_s' x_s - \mu'(x_i) x_s & -\mu_s' x_s \\ -\mu_i' x_i + \mu(x_i) & -\mu_i' x_i + \mu'(x_i) x_s + \mu_i - D & -\mu_i' x_i \\ -\mu_2 x_2 & -\mu_2 x_2 & -\mu_2' x_2 + \mu_2 - D \end{pmatrix}.$$

1.  The Jacobian matrix calculated at the steady-state $F_{000}$ is given by:

$$J_{000} = \begin{pmatrix} D_s - D & 0 & 0 \\ 0 & D_i - D & 0 \\ 0 & 0 & D_2 - D \end{pmatrix}.$$

$J_{000}$ admits three eigenvalues: $\lambda_1 = D_s - D > 0$, $\lambda_2 = D_i - D < 0$, and $\lambda_3 = D_2 - D > 0$. Then, the steady-state $F_{000}$ is a saddle point.

2.  The Jacobian matrix calculated at the steady-state $F_{100} = (s_{in} - \bar{s}_1, 0, 0)$ is given by:

$$J_{100} = \begin{pmatrix} -\mu_s'(\bar{s}_1)(s_{in} - \bar{s}_1) & -\mu_s'(\bar{s}_1)(s_{in} - \bar{s}_1) - \mu'(0)(s_{in} - \bar{s}_1) & -\mu_s'(\bar{s}_1)(s_{in} - \bar{s}_1) \\ 0 & \mu'(0)(s_{in} - \bar{s}_1) + \mu_i(\bar{s}_1) - D & 0 \\ 0 & 0 & \mu_2(\bar{s}_1) - D \end{pmatrix}$$

where $\mu_s$ and $\mu_2$ are expressed at $\bar{s}_1$. $J_{100}$ admits three eigenvalues: $\lambda_1 = -\mu_i'(\bar{s}_1)(s_{in} - \bar{s}_1) < 0$, $\lambda_2 = \mu'(0)(s_{in} - \bar{s}_1) + \mu_i(\bar{s}_1) - D = D(\mathcal{R}_0 - 1) > 0$, and $\lambda_3 = \mu_2(\bar{s}_1) - D < 0$. Then, the steady-state $F_{100}$ is a saddle point.

3.  The Jacobian matrix calculated at the steady-state $F_{001} = (0, 0, s_{in} - \bar{s}_2)$ is given by:

$$J_{001} = \begin{pmatrix} \mu_s(\bar{s}_2) - D & 0 & 0 \\ 0 & \mu_i(\bar{s}_2) - D & 0 \\ -\mu_2(\bar{s}_2)(s_{in} - \bar{s}_2) & -D(s_{in} - \bar{s}_2) & -\mu_2'(\bar{s}_2)(s_{in} - \bar{s}_2) \end{pmatrix}$$

where $\mu_s$ and $\mu_2$ are expressed at $\bar{s}_2$. $J_{001}$ admits three eigenvalues: $\lambda_1 = \mu_s(\bar{s}_2) - D > 0$, $\lambda_2 = \mu_i(\bar{s}_2) - D < 0$, and $\lambda_3 = -\mu_2'(\bar{s}_2)(s_{in} - \bar{s}_2) < 0$. Thus, the steady-state $F_{001}$ is a saddle point.

4.  The Jacobian matrix calculated at the steady-state $F_{111} = (\check{x}_s, \check{x}_i, \check{x}_2)$ is given by:

$$
J_{111} = \begin{pmatrix}
-\mu_s'\check{x}_s & -(\mu_s'\check{x}_s + \mu'(x_i)\check{x}_s) & -\mu_s'\check{x}_s \\
-\mu_i'\check{x}_i + \mu(\check{x}_i) & -\left(\mu_i'\check{x}_i - \mu'(\check{x}_i)\check{x}_s + \mu(\check{x}_i)\dfrac{\check{x}_s}{\check{x}_i}\right) & -\mu_i'\check{x}_i \\
-D\check{x}_2 & -D\check{x}_2 & -\mu_2'\check{x}_2
\end{pmatrix}
$$

where $\mu_s$, $\mu_i$, and $\mu_2$ are expressed at $(s_{in} - \check{x}_s - \check{x}_i - \check{x}_2)$.

$$
P(\lambda) = \begin{vmatrix}
-(\lambda + \mu_s'\check{x}_s) & -(\mu_s'\check{x}_s + \mu'(x_i)\check{x}_s) & -\mu_s'\check{x}_s \\
-\mu_i'\check{x}_i + \mu(\check{x}_i) & -\left(\lambda + \mu_i'\check{x}_i - \mu'(\check{x}_i)\check{x}_s + \mu(\check{x}_i)\dfrac{\check{x}_s}{\check{x}_i}\right) & -\mu_i'\check{x}_i \\
-D\check{x}_2 & -D\check{x}_2 & -(\lambda + \mu_2'\check{x}_2)
\end{vmatrix}.
$$

$J_{111}$ admits three eigenvalues: $\lambda_1$, $\lambda_2$, and $\lambda_3$, roots of the characteristic polynomial given by:

$$
\lambda^3 + a_2\lambda^2 + a_1\lambda + a_0 = 0
$$

with:

$$
\begin{aligned}
a_0 &= \mu_s'\check{x}_s\left[\mu_2'\check{x}_2\left(\mu_i'\check{x}_i - \mu'(\check{x}_i)\check{x}_s + \mu(\check{x}_i)\dfrac{\check{x}_s}{\check{x}_i}\right) - D\check{x}_2\mu_i'\check{x}_i\right] \\
&\quad - (\mu_i'\check{x}_i - \mu(\check{x}_i))[\mu_2'\check{x}_2(\mu_s'\check{x}_s + \mu'(x_i)\check{x}_s) - D\check{x}_2\mu_s'\check{x}_s] \\
&\quad + D\check{x}_2\left[\mu_i'\check{x}_i(\mu_s'\check{x}_s + \mu'(x_i)\check{x}_s) - \mu_s'\check{x}_s\left(\mu_i'\check{x}_i - \mu'(\check{x}_i)\check{x}_s + \mu(\check{x}_i)\dfrac{\check{x}_s}{\check{x}_i}\right)\right] \\
&= \mu_s'\check{x}_s\mu_2'\check{x}_2\left(\dfrac{\mu(\check{x}_i)}{\check{x}_i} - \mu'(\check{x}_i)\right)\check{x}_s\mu_2'\check{x}_2\mu_i'\check{x}_i + \mu(\check{x}_i)\mu_2'\check{x}_2\mu_s'\check{x}_s + \mu(\check{x}_i)\mu_2'\check{x}_2\mu'(x_i)\check{x}_s \\
&\quad + D\check{x}_2\mu_i'\check{x}_i\mu'(x_i)\check{x}_s + D\check{x}_2\mu_s'\check{x}_s\mu'(\check{x}_i)\check{x}_s - \mu_i'\check{x}_i\mu_2'\check{x}_2\mu'(x_i)\check{x}_s - D\check{x}_2\mu(\check{x}_i)\mu_s'\check{x}_s \\
&\quad - D\check{x}_2\mu_s'\check{x}_s\mu(\check{x}_i)\dfrac{\check{x}_s}{\check{x}_i}, \\
a_1 &= \mu_s'\check{x}_s\mu_i'\check{x}_i + \mu_s'\check{x}_s^2\left(\dfrac{\mu(\check{x}_i)}{\check{x}_i} - \mu'(\check{x}_i)\right) + \mu_s'\check{x}_s\mu_2'\check{x}_2 + \mu_2'\check{x}_2\mu_i'\check{x}_i \\
&\quad + \mu_2'\check{x}_2\left(\dfrac{\mu(\check{x}_i)}{\check{x}_i} - \mu'(\check{x}_i)\right)\check{x}_s - D\check{x}_2\mu_i'\check{x}_i - \mu_i'\check{x}_i\mu_s'\check{x}_s - \mu_i'\check{x}_i\mu'(x_i)\check{x}_s + \mu(\check{x}_i)\mu_s'\check{x}_s \\
&\quad + \mu(\check{x}_i)\mu'(x_i)\check{x}_s - D\check{x}_2\mu_s'\check{x}_s, \\
a_2 &= \mu_s'\check{x}_s + \mu_i'\check{x}_i - \mu'(\check{x}_i)\check{x}_s + \mu(\check{x}_i)\dfrac{\check{x}_s}{\check{x}_i} + \mu_2'\check{x}_2 \\
&= \mu_s'\check{x}_s + \mu_i'\check{x}_i + \left(\dfrac{\mu(\check{x}_i)}{\check{x}_i} - \mu'(\check{x}_i)\right)\check{x}_s + \mu_2'\check{x}_2.
\end{aligned}
$$

We can verify, by using Maple, that $a_2 > 0$, $a_1 > 0$, $a_0 > 0$, and $a_2a_1 > a_0$. Then, the steady-state $F_{111}$ is locally asymptotically stable once it exists.

According to Assumptions 1–3, we resume the local stability of equilibrium points in the following proposition.

**Proposition 4.** *$F_{000}$, $F_{100}$, and $F_{001}$ are saddle points; however, $F_{111}$ is stable node.*

*4.2. No Periodic Orbits on the Faces*

We start by excluding the possibility of periodic trajectory in one of the faces of the invariant set $\Lambda$.

- Consider a trajectory of dynamics (7) on the part of $\Lambda$ where $x_2 = 0$:

$$\begin{cases} \dot{x}_s &= \mu_s(s_{in} - x_s - x_i)x_s - Dx_s - \mu(x_i)x_s, \\ \dot{x}_i &= \mu_i(s_{in} - x_s - x_i)x_i - Dx_i + \mu(x_i)x_s. \end{cases} \tag{16}$$

defined on $\Lambda_{x_s x_i}$, given by:

$$\Lambda_{x_s x_i} = \left\{ (x_s, x_i) \in \mathbb{R}_+^2 : \quad x_s + x_i \le s_{in} \right\}.$$

Note that the axes $x_s = 0$ and $x_i = 0$ are invariant. Let us apply the transformation $\eta_s = \ln(x_s)$ and $\eta_i = \ln(x_i)$ for $x_s, x_i > 0$. Then, one gets the following new system:

$$\begin{cases} \dot{\eta}_s = & g_s(\eta_s, \eta_i) := \mu_s(s_{in} - e^{\eta_s} - e^{\eta_i}) - D - \mu(e^{\eta_i}), \\ \dot{\eta}_i = & g_i(\eta_s, \eta_i) := \mu_i(s_{in} - e^{\eta_s} - e^{\eta_i}) - D + \mu(e^{\eta_i})e^{\eta_s - \eta_i}. \end{cases} \tag{17}$$

Note that using Lemma 2, we have:

$$\begin{aligned} \frac{\partial g_s}{\partial \eta_s} + \frac{\partial g_i}{\partial \eta_i} &= -e^{\eta_s}\mu_s'(s_{in} - e^{\eta_s} - e^{\eta_i}) - e^{\eta_i}\mu_i'(s_{in} - e^{\eta_s} - e^{\eta_i}) \\ &\quad + e^{\eta_i}\mu'(e^{\eta_i})e^{\eta_s - \eta_i} - \mu(e^{\eta_i})e^{\eta_s - \eta_i} \\ &= -e^{\eta_s}\mu_s'(s_{in} - e^{\eta_s} - e^{\eta_i}) - e^{\eta_i}\mu_i'(s_{in} - e^{\eta_s} - e^{\eta_i}) \\ &\quad + e^{\eta_s}\left(e^{\eta_i}\mu'(e^{\eta_i}) - \mu(e^{\eta_i})\right) < 0. \end{aligned} \tag{18}$$

By the criterion of Dulac [38], system (17) (and then, system (16)) has no periodic solution. Therefore, system (7) has no periodic solution in $x_s x_i$-face ($x_2 = 0$).
- Consider a trajectory of dynamics (7) on the part of $\Lambda$ where $x_s = 0$:

$$\begin{cases} \dot{x}_i &= \mu_i(s_{in} - x_i - x_2)x_i - Dx_i, \\ \dot{x}_2 &= \mu_2(s_{in} - x_i - x_2)x_2 - Dx_2. \end{cases} \tag{19}$$

defined on $\Lambda_{x_i x_2}$, given by:

$$\Lambda_{x_i x_2} = \left\{ (x_i, x_2) \in \mathbb{R}_+^2 : \quad x_i + x_2 \le s_{in} \right\}.$$

Note that the axes $x_i = 0$ and $x_2 = 0$ are invariant. Let ua apply the transformation $\eta_i = \ln(x_i)$ and $\eta_2 = \ln(x_2)$ for $x_i, x_2 > 0$. Then, one gets the following new system:

$$\begin{cases} \dot{\eta}_i = & g_i(\eta_i, \eta_2) := \mu_i(s_{in} - e^{\eta_i} - e^{\eta_2}) - D, \\ \dot{\eta}_2 = & g_2(\eta_1, \eta_3) := \mu_2(s_{in} - e^{\eta_i} - e^{\eta_2}) - D. \end{cases} \tag{20}$$

Note that:

$$\frac{\partial g_i}{\partial \eta_i} + \frac{\partial g_2}{\partial \eta_2} = -e^{\eta_i}\mu_i'(s_{in} - e^{\eta_i} - e^{\eta_2}) - e^{\eta_2}\mu_2'(s_{in} - e^{\eta_i} - e^{\eta_2}) < 0. \tag{21}$$

From the Dulac criterion [38], system (20) (and then, system (19)) has no periodic solution. Therefore, system (7) has no periodic solution in $x_i x_2$-face ($x_s = 0$).
- Consider a trajectory of dynamics (7) on the part of $\Lambda$ where $x_i = 0$:

$$\begin{cases} \dot{x}_s &= \mu_s(s_{in} - x_s - x_2)x_s - Dx_s, \\ \dot{x}_2 &= \mu_2(s_{in} - x_s - x_2)x_2 - Dx_2. \end{cases} \tag{22}$$

defined on $\Lambda_{x_s x_2}$, given by:

$$\Lambda_{x_s x_2} = \left\{ (x_s, x_2) \in \mathbb{R}_+^2 : \quad x_s + x_2 \le s_{in} \right\}.$$

Note that the axes $x_s = 0$ and $x_2 = 0$ are invariant. Let ua apply the transformation $\eta_1^s = \ln(x_s)$ and $\eta_2 = \ln(x_2)$ for $x_s, x_2 > 0$. Then, one gets the following new system:

$$\begin{cases} \dot{\eta}_s = & g_s(\eta_s, \eta_2) := \mu_s(s_{in} - e^{\eta_s} - e^{\eta_2}) - D, \\ \dot{\eta}_2 = & g_2(\eta_2, \eta_3) := \mu_2(s_{in} - e^{\eta_s} - e^{\eta_2}) - D. \end{cases} \tag{23}$$

Note that:

$$\frac{\partial g_s}{\partial \eta_s} + \frac{\partial g_2}{\partial \eta_2} = -e^{\eta_s}\mu_s'(s_{in} - e^{\eta_s} - e^{\eta_2}) - e^{\eta_2}\mu_2'(s_{in} - e^{\eta_s} - e^{\eta_2}) < 0. \tag{24}$$

By the criterion of Dulac [38], system (23) (and then, system (22)) has no periodic solution. Therefore, system (7) has no periodic solution in $x_s x_2$-face ($x_i = 0$).

### 4.3. Persistence

In this subsection, we aim to prove the coexistence of both species 1 (either infected or not) and species 2 by proving the uniform persistence of dynamics (7). The saddle points $F_{000}$, $F_{100}$, and $F_{001}$ are the only boundary steady states for the dynamics (7). Then, we apply the proof used in [37,43,44] using the Butler-McGehee Lemma [38] frequently to prove the persistence of system (7).

**Theorem 1.** *Dynamics (7) is persistent.*

**Proof.** All the faces $x_s x_i$, $x_s x_2$, and $x_i x_2$ are invariant. Furthermore, stable and unstable manifolds of the boundary equilibrium points are represented in Figure 4.

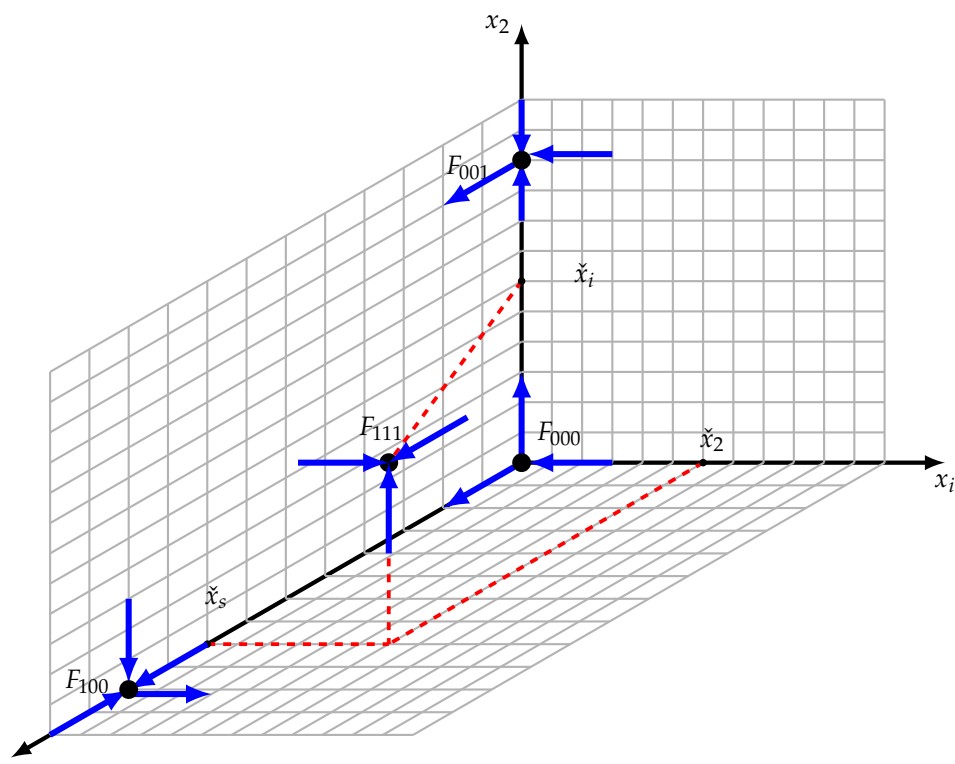

**Figure 4.** Equilibria configuration. $F_{000}$, $F_{100}$, and $F_{001}$ are saddle points; however, $F_{111}$ is an asymptotically stable interior equilibrium.

Consider a solution $\vec{z} = (x_s(t), x_i(t), x_2(t))$ with an initial condition $\vec{z}(0) = (x_s(0), x_i(0), x_2(0))$, where $x_s(0) > 0$, $x_i(0) > 0$, and $x_2(0) > 0$ are given data. Let us denote $\omega = \omega(\gamma^+(\vec{z}(0)))$ to be the omega limit set of $\gamma^+(\vec{z}(0))$, where $\gamma^+(\vec{z}(0))$ is the positive

semi-orbit passing through $\vec{z}(0)$. We aim to prove that the omega limit set has no points on each of the three faces.

- Assume that $F_{000} \in \omega$. Then, $\exists\, z^* \neq F_{000}$ inside $\omega \cap W^s(F_{000})\backslash\{F_{000}\}$. The stable manifold $W^s(F_{000})$ is one-dimensional and is restricted to the $x_i$-axis. Therefore, the entire orbit passing through $z^*$, which is inside $\omega$, becomes unbounded, which contradicts the existence of $z^*$.

- Assume that $F_{100} \in \omega$. $F_{100}$ is a saddle point with a stable manifold, $W^s(F_{100})$, of dimension two, restricted to the $x_s x_2$-plane. Therefore, $\{F_{100}\}$ is not the entire omega limit set $\omega$. Using the Butler-McGehee Lemma [38], there exists a point $z^* \neq F_{100}$ inside $\omega \cap W^s(F_{100})\backslash\{F_{100}\}$. Since $W^s(F_{100})$ lies entirely in the $x_s x_2$-plane, and since the entire orbit through $z^*$ is in $\omega$, this orbit is unbounded, which contradicts the fact that $F_{100}$ is inside $\omega$.

- Assume that $F_{001} \in \omega$. Since $F_{001}$ is a saddle point where its stable manifold $W^s(F_{001})$ is of dimension two and is restricted to the $x_i x_2$-plane, then $\{F_{001}\}$ is not the entire omega limit set $\omega$. Therefore, using the Butler-McGehee Lemma [38], there exists a point $z^* \neq F_{001}$ inside $\omega \cap W^s(F_{001})\backslash\{F_{001}\}$. Since $W^s(F_{001})$ lies entirely in the $x_i x_2$-plane, and since the entire orbit through $z^*$ is in $\omega$, this orbit is unbounded, which contradicts the fact that $F_{001}$ is inside $\omega$.

Now, let $\underline{z} = (\underline{x}_s(t), \underline{x}_i(t), \underline{x}_2(t))$ with at least one of the components $\underline{x}_s(t)$, $\underline{x}_i(t)$, and $\underline{x}_2(t)$ is zero, and suppose that $\underline{z} \in \omega$. Thus, the entire orbit passing through $\underline{z}$ should be inside $\omega$. However, since the orbit should lie entirely inside either $x_s x_i$, $x_i x_2$, or $x_s x_2$ faces, it should converge to one of the boundary equilibrium points, since there is no periodic trajectory. Therefore, this boundary equilibrium point is inside $\omega$, which contradicts the fact that all boundary equilibrium points are saddle points. Therefore, each of the components of the trajectory is greater than zero:

$$\liminf_{t \to \infty} x_s(t) > 0, \; \liminf_{t \to \infty} x_i(t) > 0 \; \text{ and } \; \liminf_{t \to \infty} x_2(t) > 0,$$

and then system (7) is persistent (see Section 4.3 in [44] for another example). □

### 4.4. Uniform Persistence of System (7)

Persistence and uniform persistence [45] are equivalent in many examples of mathematical models. Recall a theory in [45] stating that if $\mathcal{D}$ is a dynamical system, such that $\mathbb{R}^3_+$ and $\partial\mathbb{R}^3_+$ are both invariant, then $\mathcal{D}$ is uniformly persistent if it satisfies the following statements.

1. $\mathcal{D}$ is weakly persistent;
2. $\mathcal{D}$ is dissipative;
3. $\partial\mathcal{D}$ is isolated, where $\partial\mathcal{D}$ be the restriction of $\mathcal{D}$ to $\partial\mathbb{R}^3_+$;
4. $\partial\mathcal{D}$ is acyclic.

Consider the dynamics $\mathcal{D}$ on the invariant attractor bounded set $\Sigma$. We can apply the theorem given in [45] if $\partial\Sigma = \Sigma_1 \cup \Sigma_2$ and $\mathcal{D}$ is invariant on $\Sigma_1$, but repelling into the interior of $\Sigma$ on $\Sigma_2$ if Conditions 3 and 4 are satisfied when restricting $\mathcal{D}$ to $\Sigma_1$. It is clear that condition 1 is satisfied. Condition 2 is also satisfied according to Theorem 1. Condition 3 is satisfied because all boundary equilibrium points are hyperbolic and then their union forms a covering of the omega limit sets of $\Sigma_1$. Condition 4 is also satisfied because the boundary equilibrium points are not linked cyclically. Thus, we conclude on the uniform persistence of system (7).

**Theorem 2.** *Dynamics (7) is uniformly persistent, i.e.,* $\exists\, \beta > 0$, *such that:*

$$\liminf_{t \to \infty} x_s(t) > \beta, \liminf_{t \to \infty} x_i(t) > \beta, \liminf_{t \to \infty} x_2(t) > \beta.$$

## 5. Uniform Persistence of System (2)

Return to the main mathematical model (2) describing the competition of two bacteria in a chemostat in the presence of a virus that affects only the first bacteria. System (2) admits $E_{000} = (s_{in}, 0, 0, 0)$, $E_{100} = (\bar{s}_1, s_{in} - \bar{s}_1, 0, 0)$, $E_{001} = (\bar{s}_2, 0, 0, s_{in} - \bar{s}_2)$, and $E_{111} = (s_{in} - \check{x}_s - \check{x}_i - \check{x}_2, \check{x}_s, \check{x}_i, \check{x}_2)$ as equilibrium points. $E_{000}$, $E_{100}$, and $E_{001}$ are saddle points; however, $E_{111}$ is locally asymptotically stable. We need to prove the uniform persistence of the main system (2). Let $\vec{z_0} = (s(0), x_s(0), x_i(0), x_2(0))$ with $s(0) \geq 0$, $x_s(0) \geq 0$, $x_i(0) \geq 0$ and $x_2(0) \geq 0$; then, $\omega(\vec{z_0}) \in \Sigma$. Furthermore, assume that $\exists \vec{\sigma} \in \mathbb{R}_+^4 \setminus \Sigma$ such that the the solution converges to $\vec{\sigma}$. This is not possible since $\Sigma$ is a global attractor according to proposition 1. Now, suppose that $\omega(\vec{z_0})$ contains a point on one of the faces where one of the variables $x_s$, $x_i$, or $x_2$ is zero; therefore, the entire trajectory passing through this point should be inside $\omega(\vec{z_0})$. Thus, the omega limit set $\omega(\vec{z_0})$ should be entirely inside $\Sigma$.

**Theorem 3.** *Dynamics (2) is uniformly persistent, i.e.,* $\exists \eta > 0$*, such that:*

$$\liminf_{t \to \infty} s(t) > \eta, \liminf_{t \to \infty} x_s(t) > \eta, \liminf_{t \to \infty} x_i(t) > \eta, \liminf_{t \to \infty} x_2(t) > \eta.$$

## 6. Numerical Simulations

We cofirm the theoretical findings by some numerical results using Monod functions (or Holling's functions type II) to express all growth rates and the incidence rate:

$$\mu_s(s) = \frac{\bar{\mu}_s s}{k_s + s}, \mu_i(s) = \frac{\bar{\mu}_i s}{k_i + s}, \mu_2(s) = \frac{\bar{\mu}_2 s}{k_2 + s} \text{ and } \mu(s) = \frac{\bar{\mu} s}{k + s}$$

where $k_s, k_i, k_2$, and $k$ are Monod constants. $\bar{\mu}_s, \bar{\mu}_i, \bar{\mu}_2$, and $\bar{\mu}$ are positive constants. We used Holling type-II functions as typical examples [46,47] since they are nonlinear and satisfied all our assumptions on growth rates and incidence rates. All constants are chosen such that the functions $\mu_s, \mu_i, \mu_2$, and $\mu$ satisfy Assumptions 1–3.

Consider the parameters values given in the following Table 2.

**Table 2.** The parameter values are used to illustrate the theoretical results, but they have no biological significance.

| Parameter | $\bar{\mu}_s$ | $\bar{\mu}_i$ | $\bar{\mu}_2$ | $\bar{\mu}$ | $k_s$ | $k_i$ | $k_2$ | $k$ | $s_{in}$ | $D_s$ | $D_2$ | $D_i$ |
|---|---|---|---|---|---|---|---|---|---|---|---|---|
| Value | 3 | 1 | 2.4 | 3 | 3 | 2.5 | 5 | 2 | 10 | 2.3 | 1.6 | 0.8 |

We give two examples that satisfy Assumptions 1–3, which ensures the persistence of both species 1 (either infected or not) and species 2, as seen in Figures 5 and 6.

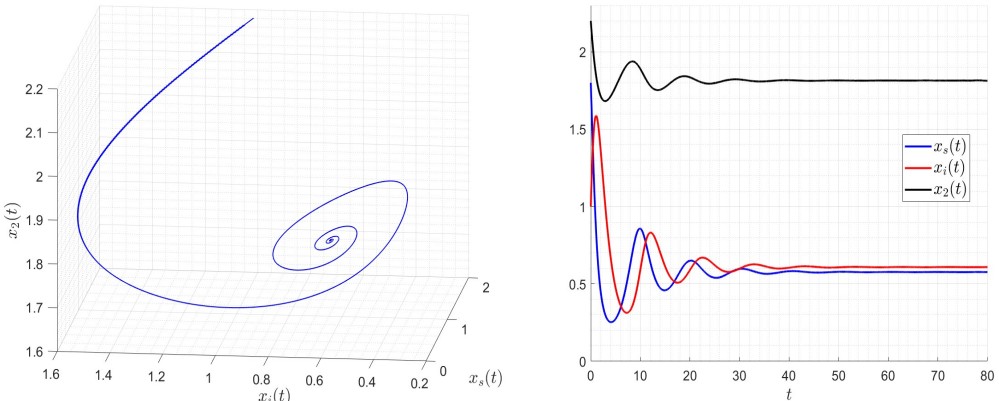

**Figure 5.** $D = 1.4$ and $\mathcal{R}_0 = 8.27$ and then Assumptions 1–3 are satisfied. The solution of the system (2) converges to the equilibrium $F_{111}$ where the two species coexist (either infected or not).

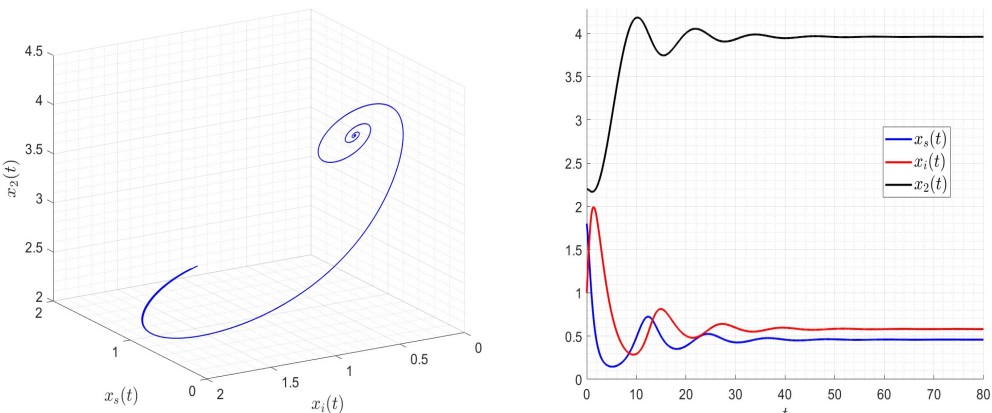

**Figure 6.** $D = 1.2$, $\mathcal{R}_0 = 10.37$ and then Assumptions 1–3 are satisfied. The solution of the system (2) converges to the equilibrium $F_{111}$ where the two species coexist (either infected or not).

## 7. Conclusions

Since we aim to prove that the competitive exclusion principle is not usually valid when two competitors grow on a single essential resource, we add, in this paper, an additional mechanism of competition by adding a virus in the chemostat that affects only the first species, and then the coexistence becomes possible. We propose a mathematical model that describes the competition of two species in a chemostat in the presence of a virus. We suppose that only one population is affected by the virus. We suppose also that the substrate is continuously added to the bioreactor. We obtain a model taking the form of an 'SI' epidemic model. The stability of the steady states was carried out. The system can have multiple steady states with which we can determine the necessary and sufficient conditions for both existence and local stability. We exclude the possibility of periodic orbits and we prove the uniform persistence of both species. Finally, we give some numerical simulations that validate the obtained results.

The main result of this work is that the presence of the virus allows the coexistence of the two bacterial species when the species cannot coexist unless the virus is present. A biological explanation of this result is that the virus affects the species which should win the competition and then it gives the opportunity to the second species to persist.

**Author Contributions:** Conceptualization, M.E.H. and A.H.A.; methodology, M.E.H. and A.H.A.; writing—original draft, M.E.H. and A.H.A.; writing—review and editing, M.E.H. and A.H.A. All authors have read and agreed to the published version of the manuscript.

**Funding:** This research work was funded by Institutional Fund Projects under grant no. IFPIP: 672-130-1443.

**Data Availability Statement:** Not applicable.

**Acknowledgments:** This research work was funded by Institutional Fund Projects under grant no. IFPIP: 672-130-1443. The authors gratefully acknowledge the technical and financial support provided by the Ministry of Education and King Abdulaziz University, DSR, Jeddah, Saudi Arabia. The authors are also grateful to the unknown referees for the many constructive suggestions, which helped to improve the presentation of the paper.

**Conflicts of Interest:** The authors declare no conflict of interest.

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
