# Peer review of "Bacterial Competition in the Presence of a Virus in a Chemostat"

_mathematics, doi:10.3390/math11163530_

Round 1

Reviewer 1 Report

Authors have derived a mathematical model that describes the competition of two populations in a chemostat in the presence of a virus where only one population is affected by the virus. Supposing that the substrate is continuously added to the bioreactor, they have obtained a model taking the form of an ’SI’ epidemic model using general increasing growth rates of bacteria on the substrate and a general increasing incidence rate for the viral infection. The stability of the steady states was carried out. Necessary and sufficient conditions for both existence and local stability are examined. Numerical simulations are performed to validate the obtained results. The analysis presented here is interesting. However, authors need to take care of the following.

1.      The source of Fig. 1 must be cited.

2.      Entire manuscript should be examined for grammatical and typo errors. Moreover, authors are suggested to take care of ‘articles’.

3.      In this work, the authors have investigated a 4D model after converting it to a 3D model with a generalized functional response. In numerical simulations, they have considered Holling type-II functional response, which is already extensively studied in the literature. Only the species are different. Please highlight the novelty of the present work.

4.      In numerical simulation, authors need to explain the biological meanings of the findings in the Figures. Figures need to be discussed in detail relating them to theoretical findings.

5.      Fig. 5 and Fig. 7 both represent coexistence. Then what is the difference in interpretation?  Please show the simulations for other theoretical findings too.

6.      Conclusion should be different from the abstract. The conclusion should contain the findings of the study.

 Please highlight the changes made by different color text.

 Moderate editing of English language required

Author Response

Ref: mathematics-2559445.

Title: Bacterial Competition in the Presence of a Virus in a Chemostat.

Journal: Mathematics.

Thank you for the opportunity to revise our manuscript. We appreciate the careful review and constructive suggestions and comments. We have revised the manuscript accordingly and provide specific answers below.

1. The source of Fig. 1 must be cited.   

  • A reference was added to the caption of figure 1.

2. Entire manuscript should be examined for grammatical and typo errors. Moreover, authors are suggested to take care of ‘articles’.

  • We have done thorough English editing and corrected some grammatical mistakes in the revised manuscript. Some references were either modified or added and discussed inside the text.

3. In this work, the authors have investigated a 4D model after converting it to a 3D model with a generalized functional response. In numerical simulations, they have considered Holling type-II functional response, which is already extensively studied in the literature. Only the species are different. Please highlight the novelty of the present work.

  • We used Holling type-II functions as typical examples [48,49] since they are nonlinear and satisfied all our assumptions on growth rates and incidence rate.

4. In numerical simulation, authors need to explain the biological meanings of the findings in the Figures. Figures need to be discussed in detail relating them to theoretical findings.

  • The figures are discussed by giving more biological meaning of the behaviors for the captions.

5. Fig. 5 and Fig. 7 both represent coexistence. Then what is the difference in interpretation?  Please show the simulations for other theoretical findings too.

  • We just gave two examples that satisfy the Assumptions 1, 2 and 3 which ensure the persistence of both species. Please note that the theoretical study considers only assumptions that ensure the coexistence of both, species 1 (either infected or not) and species 2.

6. Conclusion should be different from the abstract. The conclusion should contain the findings of the study.

  • The conclusion was modified to reflect the findings of this study.

Reviewer 2 Report

This work concerns on a bacterial competition of two species in the presence of a virus in a chemostat. S appeared in Eq. (1) as well as Fig.2 denotes the concentration of the resource. Amid the two species, species 1 is present in two compartments, susceptible (Xs) and infected (Xi). But, species 2 is present in a single form (X2). The growth rate of the two species is accounted (by Mu_{s,i,2}). Infection flux is accounted by f(X_i)*X_s obeying to SI process. D implies the flow rate in the chemostat, i.e., a bio-reactor as shown in Fig. 1.

Taking a normalization, the authors finalized the ODEs as in Eq. (2).

Following to the model building, the authors explored the theoretical analysis of the model above. They got R_0. And they read down the original ODEs to simple 3D dynamics represented by Eq. (7), which allows four equilibria; F_000, F_100, F_001, and F_111. They analyze the stability of those. Subsequently, they delivered some numerical results.

Although what the authors can successfully bring by the present MS seems not so much in terms of stringent scientific viewpoint, because their model seems specific, their model building, mathematical healthiness for their analysis, and the presentation of numerical results are fine, and somehow informative to the audience. Hence, I have a relatively positive evaluation of this work.

One suggestion is as below. Their model like the SI-diffusion-reaction process should be more carefully explained to the audience. They should review some epidemiological models of not only SI but also SIS, SIR(S), SEIR(S) and others by referring to some good review papers and books such as Sociophysics Approach to Epidemics, Springer, 2021, simply because all of the audience might be very much interested in the epidemiological model by ODEs in the wake of COVID-19.

Author Response

Ref: mathematics-2559445.

Title: Bacterial Competition in the Presence of a Virus in a Chemostat.

Journal: Mathematics.

Thank you for the opportunity to revise our manuscript. We appreciate the careful review and constructive suggestions and comments. We have revised the manuscript accordingly and provide specific answers below.

One suggestion is as below. Their model like the SI-diffusion-reaction process should be more carefully explained to the audience. They should review some epidemiological models of not only SI but also SIS, SIR(S), SEIR(S) and others by referring to some good review papers and books such as Socio-physics Approach to Epidemics, Springer, 2021, simply because all of the audience might be very much interested in the epidemiological model by ODEs in the wake of COVID-19.

  • The manuscript was rewritten as per reviewer’s suggestions and the proposed mathematical model was more explained. Some references related to some epidemiological models such as SIS, SIR, SEIR, SEIRS, SVEIR were added and discussed inside the text.
  • We have done thorough English editing and corrected some grammatical mistakes in the revised manuscript.

Reviewer 3 Report

The paper presents a mathematical model of bacterial competition in the presence of a virus in a chemostat. The authors propose a generalized model that takes into account the growth rates of bacteria on the substrate and the incidence rate of viral infection. The model shows that the system is uniformly persistent, meaning that the bacterial populations remain bounded away from extinction over an infinite time horizon. Overall, the paper is well written, but the authors should proofread the paper for typographical errors, e.g., line 72 should read "know" instead of “now".

I have two main criticisms that the authors should address in their revision. First, the assumption in line 97 that \mu (x) = f(x) seems completely arbitrary and overly restrictive, since in principle there is no relation between these functions. The authors should justify this assumption and discuss its possible consequences. Second, the paper shows that the presence of the virus allows the coexistence of the two bacterial species. The authors should explain why this is the case, without relying on their mathematical results.

It is good, but can be improved with a second reading.

Author Response

Ref: mathematics-2559445.

Title: Bacterial Competition in the Presence of a Virus in a Chemostat.

Journal: Mathematics.                                           

Thank you for the opportunity to revise our manuscript. We appreciate the careful review and constructive suggestions and comments. We have revised the manuscript accordingly and provide specific answers below.

Overall, the paper is well written, but the authors should proofread the paper for typographical errors, e.g., line 72 should read "know" instead of “now".

  • The manuscript was rewritten as per reviewer’s suggestions. We have done thorough English editing and corrected the grammatical mistakes in the revised manuscript.

I have two main criticisms that the authors should address in their revision. First, the assumption in line 97 that \mu (x) = f(x) seems completely arbitrary and overly restrictive, since in principle there is no relation between these functions. The authors should justify this assumption and discuss its possible consequences.

  • Please note that \mu(x_i)=f(Y_1 x_i) (line 97 in the previous manuscript version) is simply a new notation of the incidence rate due to the change of variable. It is different than the growth rates \mu_i, \mu_s, \mu_2.

Second, the paper shows that the presence of the virus allows the coexistence of the two bacterial species. The authors should explain why this is the case, without relying on their mathematical results.

  • The main result of this work is the presence of the virus allows the coexistence of the two bacterial species when the species cannot coexist unless the virus is present. A biological explanation of this result is that the virus affects the species who should wins the competition and then it gives the opportunity to the second species to persist. We added this paragraph to the conclude remarks.

Reviewer 4 Report

The topic is actual and fits to special issue.

The paper is well-written and illustrated. Some comments should be addressed in order the work should be accepted.

Comments:

1. When analyzing the related works some words should be mentioned on global asymptotic stability and Lyapunov method.

2. Line 154. Please describe the way how you obtained the basic reproduction number (next-generation operator?).

3. When analyzing the related papers I recommend to mention DOI: 10.1109/ACCESS.2021.3104519  devoted to multistrain virus modeling.

Author Response

Ref: mathematics-2559445.

Title: Bacterial Competition in the Presence of a Virus in a Chemostat.

Journal: Mathematics.

 Thank you for the opportunity to revise our manuscript. We appreciate the careful review and constructive suggestions and comments. We have revised the manuscript accordingly and provide specific answers below.

  1. When analyzing the related works some words should be mentioned on global asymptotic stability and Lyapunov method.
    The introduction was updated to include some relevant references. We have mentioned some works that investigated the global asymptotic stability, in particular those who used Lyapunov method.
  2. Line 154. Please describe the way how you obtained the basic reproduction number (next-generation operator?).
    Please note that the basic reproduction number R_0 for system (2) was calculated using the next-generation operator approach proposed in [39] and deduced from the third equation (infected compartment) of system (2). The text was modified: please see page 6 line 153.

  3. When analyzing the related papers, I recommend to mention DOI: 10.1109/ACCESS.2021.3104519 devoted to multi-strain virus modeling.
    The reference was added and discussed inside the text.